# Chronic Kidney Disease Diets for Kidney Failure Prevention: Insights from the IL-11 Paradigm

**DOI:** 10.3390/nu16091342

**Published:** 2024-04-29

**Authors:** Denise Elshoff, Priyanka Mehta, Ouliana Ziouzenkova

**Affiliations:** 1School of Health and Rehabilitation Sciences, The Ohio State University, Columbus, OH 43210, USA; elshoff.3@osu.edu; 2Department of Human Sciences, The Ohio State University, Columbus, OH 43210, USA; mehta.487@osu.edu

**Keywords:** renal disease, renal diets, renal regeneration, lutein, IL11, renal tubule epithelial cells TEC, mesenchymal transition

## Abstract

Nearly every fifth adult in the United States and many older adults worldwide are affected by chronic kidney disease (CKD), which can progress to kidney failure requiring invasive kidney replacement therapy. In this review, we briefly examine the pathophysiology of CKD and discuss emerging mechanisms involving the physiological resolution of kidney injury by transforming growth factor beta 1 (TGFβ1) and interleukin-11 (IL-11), as well as the pathological consequences of IL-11 overproduction, which misguides repair processes, ultimately culminating in CKD. Taking these mechanisms into account, we offer an overview of the efficacy of plant-dominant dietary patterns in preventing and managing CKD, while also addressing their limitations in terms of restoring kidney function or preventing kidney failure. In conclusion, this paper outlines novel regeneration strategies aimed at developing a reno-regenerative diet to inhibit IL-11 and promote repair mechanisms in kidneys affected by CKD.

## 1. Introduction

Chronic kidney disease (CKD) is a serious public health concern affecting an estimated 14.0% of adults in the United States (Box 1) [1]. It is a progressive disease that eventually leads to end-stage renal disease (ESRD), which requires renal replacement therapy via hemodialysis, peritoneal dialysis, or kidney transplant. With limited hemodialysis or no treatment, ESRD patients often transition to palliative/hospice care, contributing to overall Medicare costs for this disease reaching USD 50.8 billion, which represents 6.1% of total Medicare expenditures [1]. ESRD is a morbid disease with low survival rates: the 5-year survival probability is 0.23 in adults aged 75 and older and 0.65 in young adults, aged 18–44 years [1]. The 5-year survival rate with ESRD is generally much lower on dialysis (children: HD, 0.81; PD, 0.86 and adults aged 65–74: HD, 0.34, PD, 0.38) than with kidney transplant (children: 0.96 and adults aged 65–74: 0.82) [1]. Although survival rates after kidney transplant are substantially higher than on dialysis, kidney transplantation is limited by donor kidney availability, risk of graft failure, and side effects of long-term immunosuppressant medications [2,3,4].

Box 1CKD definition [5,6,7,8,9]. CKD < 60 mL/min estimated glomerular flow rate (eGFR) or at least one other sign of abnormal kidney structure or functional loss, like albuminuria for more than three months.-Mild irreversible kidney damage, Stages 1–2: >60 mL/min eGFR, albuminuria.-Advanced CKD Stages 3–4: 15–59 mL/min eGFR.-ESRD, Stage 5: 15 mL/min eGFR, kidney failure, and mortal level of uremic toxins. Renal replacement therapy required to prevent death in days or weeks.


Moreover, mortality rates among kidney transplant recipients aged 66–74 exceed the mortality rates observed in individuals of the same age group with other conditions such as cancer, diabetes mellitus (DM), or stroke [1]. As the US population ages and rates of obesity and DM increase, the prevalence of CKD and ESRD is expected to rise, along with the associated costs [10,11]. 

Often asymptomatic until Stages 4–5, CKD is an insidious disease [5]. Stage 3–4 CKD affects every fourth patient with DM and every third patient with hypertension [10]. Identifying and screening for CKD facilitates earlier diagnosis and treatment, with the aim of averting the morbidity and costs associated with advanced CKD and ESRD, albeit these criteria may differ among age groups (Box 2) [12]. Therefore, management of hyperglycemia, hypertension, and obesity are critical for reducing the risk of CKD development, progression, and mortality [11,13,14,15,16].

Box 2Age-related CKD risks. Stage 3 CKD eGFR of 30–59 mL/min meets 20% of adults above 65 years old and only 1% younger than 65. Age-related eGFR reduction does not substantially increase the risk of progression to ESRD or mortality, and some propose to change the definition for CKD to eGFR of less than 45 mL/min in adults older than 65 years [17]. A total of 67 times more older CKD patients remain at Stage 3 CKD compared to those who have progressed to Stage 5 CKD. Conversely, younger individuals progress to Stage 5 CKD more rapidly, with only 12 times more young people remaining at Stage 3 than at Stage 5 CKD [1]. These trends for CKD progression are influenced by comorbidities.

Diet critically impacts the development and progression of CKD, both directly and by influencing hyperglycemia, hypertension, obesity, and inflammation [5,18,19,20,21]. Therefore, even for primary renal diseases, like polycystic kidney disease and IgA nephropathy, the management of dietary composition and regimen can slow disease progression and complications [22,23]. Patients in early CKD Stages 1–2 are recommended to adhere to a renal protective diet (Box 3) and lifestyle proposed for the general population that includes physical activity, smoking cessation, and management of weight, hyperglycemia, hypertension, and other comorbidities that includes other core elements defined in 2015 as a healthy dietary pattern by the Dietary Guidelines Advisory Committee [24].

Box 3Overview of renal diets defined as a healthy dietary pattern (HDP) by the Dietary Guidelines Advisory Committee in 2015 [24].
-HDP: high intake: fruits/vegetables. Low intake: consumption of sodium/trans and saturated fats/added sugars.-A renal protective diet: moderate protein (0.8–1.0 g/kg body weight/day).-Mediterranean style diets (MedDiet): high intake: olive oil/fruits/, vegetables/nuts/whole grains. Moderate intake: fish/lean poultry/low-fat dairy/red wine. Low intake: sweets/red meat [25,26].-Dietary Approaches to Stop Hypertension (DASH) MedDiet and low intake: sodium/sweets/full-fat dairy/fatty meats [25,27]. 


These measures may slow CKD progression to avoid or prevent advanced CKD. Several specific diets, DASH and MedDiet, have been composed to meet these criteria (Box 3). For moderate and advanced CKD Stages 3–4, personalized conservative treatment is recommended, often with a low-protein diet (0.55–080 g/kg body weight/day), HDP and lifestyle, avoidance of nephrotoxic medications, and renoprotective medications, including a renin-angiotensin system modulator (Angiotensin-converting enzyme inhibitors (ACEi) or angiotensin II Receptor Blockers (ARB) and Sodium-Glucose Co-Transporter 2 (SGLT2) inhibitor, to manage symptoms and slow CKD progression to prevent or delay ESRD [5,6,7,8,9,28]. The goal of dietary guidelines and pharmaceutical treatments is to prevent further kidney damage and delay progression to ESRD because kidney damage is considered irreversible, even in the earliest stages of CKD.

Although progressive kidney damage is typically considered irreversible, interventions such as simultaneous pancreas and kidney transplants in T1DM patients [29], as well as weight loss in obese adults [30] and a child [31], have demonstrated potential to reverse CKD. Research on the role of mesenchymal stem cells in adaptive repair and a more complete understanding of the mechanisms behind maladaptive damage leading to CKD has the potential to transform future treatments, including dietary interventions. 

In this review, we will briefly examine the pathophysiology of CKD, with a focus on recently proposed mechanisms for adaptive kidney regeneration involving the TGFβ/IL11/Wnt repair secretome, alongside a maladaptive response hindered by interleukin 11 (IL-11) overproduction, ultimately resulting in progressive and irreversible kidney damage [32,33,34]. Considering these mechanisms, we delve into the effectiveness and constraints of nutritional interventions based on plant-dominant dietary patterns for the prevention and management of CKD. Finally, we provide an overview of nutrient suppressors of IL-11 and their association with CKD, with the aim of informing researchers and clinicians about emerging mechanisms in both regenerative and nutritional science. These insights may inspire and rationalize future studies aimed at advancing interventions from kidney protection to kidney regeneration.

## 2. Acute and Chronic Triggers of CKD

CKD originates from an acute or chronic etiological trigger that initiates kidney injury, followed by a perpetuating mechanism that sustains damage in the pathogenic process. This consensus was achieved through a comprehensive analysis of integrated systemic processes and the role of malnutrition in CKD pathogenesis, conducted by experts from the European Renal and Cardiovascular Medicine Working Group of the European Renal Association and European Dialysis Transplantation Association (ERA-EDTA) [5,35].

Acute kidney injury (AKI) causes severe damage in a short period, whereas a chronic trigger gradually damages the kidney over months or years. The etiology of AKI may be prerenal, intrinsic, or post-renal [36]. Prerenal AKI is caused by a condition that decreases blood flow to the kidneys and leads to hypoxia, such as hypotension, hypovolemic or anaphylactic shock, severe burns, or a sudden decrease in cardiac output due to an ischemic event (Figure 1).

Intrinsic AKI is caused by direct damage to the kidney from sepsis, trauma, or nephrotoxic drugs. Post-renal AKI occurs due to urinary obstruction caused by factors such as kidney stones, blood clots, tumors, scarring, or other structures, resulting in congestion within the renal filtration system [36]. Chronic triggers may also be prerenal (e.g., decreased cardiac output due to heart failure), intrinsic (e.g., polycystic kidney disease, primary glomerulonephritis), or post-renal (e.g., prostate disease). In the USA, DM, hypertension, and cardiovascular disease are the most common chronic triggers of CKD. Other contributing factors include aging (age 65 years and older), overweight or obesity, black race, family history of kidney disease, and history of smoking or use of nephrotoxic medications [1,5,37]. 

The interaction between injured TECs and immune cells is triggered by severe or recurrent kidney injuries, leading to the release of various factors such as cytokines, miRNA, and growth factors, including transforming growth factor β1 (TGFβ1). These factors, in turn, influence stem cells, reparative fibrosis by myofibroblasts, and inflammation, all of which are pivotal in promoting adaptive kidney repair and regeneration [5] (Figure 1). However, hypoxic conditions resulting in mitochondrial dysfunction can impede the cessation of ECM production and the resolution of local and systemic inflammation processes, leading to their transition into a chronic and detrimental state [38], including glomerulosclerosis and tubulointerstitial fibrosis [5,39]. The persistent injury stems from a self-perpetuating mechanism driven by the partial epithelial-mesenchymal transition (pEMT) of injured TECs into tubular epithelial cells with a mesenchymal phenotype (pTECs). These pTECs express vascular adhesion molecule 1 (VCAM1+), facilitating interaction with immune cells [40]. Recent studies have underscored the significance of the pTEC state [34], which can either lead to the repopulation of TECs and extracellular matrix (ECM) with a subsequent repair or perpetuate a progressive cycle of kidney damage [38], characterized by glomerular hyperfiltration and microvascular dysfunction [5,41]. The fate of pTECs is dependent on extracellular signals initiated by TGFβ1 and orchestrated by IL-11.

## 3. The Discrete Roles of TGFβ1 and IL-11 in Adaptive and Maladaptive Repair of Kidney Damage

Strategic positioning of latent TGFβ, including renal TGFβ, within the extracellular matrix (ECM) makes them primary pathways responding to acute or chronic injury through activation of the canonical transcription factor complex SMAD2/3 and SMAD4. This pathway and its major outcomes, including the expression and secretion of IL-11 and Wnt, are depicted in Figure 2 [42,43]. Noncanonical activation of the same oligomeric complex by TGFβ1 proceeds with the release of activated TAK1, subsequently inducing inflammatory, cell cycle, and differentiation pathways. TGFβ pathway activation also occurs in fibroblasts or myofibroblasts, stimulating collagen expression through noncanonical TAK1 pathways and consequently enhancing their fibrotic response [43]. The initial role of TGFβ1 as a “first responder” led researchers to assume its involvement as a mediator of kidney damage. However, clinical attempts to utilize anti-TGFβ1 therapies to treat CKD resulted in exacerbated kidney inflammation and damage [34,44]. Additional research is necessary to ascertain whether anti-TGFβ1 therapy causes toxicity by merely blocking specific anti-inflammatory aspects of TGFβ1 signaling [45] or fundamentally by impeding renoprotective latent TGFβ1 [34]. It is also conceivable that this therapy inhibits miRNA-133a, which in turn induces antifibrotic and reparative pathways activated by TGFβ1 signaling [46], as observed in the context of post-myocardial infarction [47]. Despite the absence of a complete mechanism, these results suggest a role for TGFβ1 in repair processes and highlight a repair secretome containing IL-11 and Wnt as molecules that determine the transition from the physiological to pathological outcomes of this process.

The cytokine IL-11, expressed in response to TGFβ1/SMAD activation, seems to play a pivotal role in the transition from adaptive repair processes to maladaptive damaging processes in CKD in response to kidney insult, whether it stems from AKI or a chronic disease such as diabetes [34]. The signaling processes of secreted IL-11 and WNT are interdependent and involve canonical or alternative pathways induced by IL-11.

In a physiological setting, IL-11 forms a signaling complex in which this cytokine binds to its cognate IL-11 receptor subset alpha (IL-11Rα) and gp130 receptor (Figure 3) [49], similar to other members of the IL-6 family of cytokines (reviewed in [50]). Unlike the IL-6 receptor, IL-11Rα is expressed on epithelial/polarized cells, like kidney TECs as well as stromal cells, like fibroblasts and smooth muscle cells [33,51]. Binding of IL-11 to the IL-11Rα/gp130 receptor complex recruits and auto-phosphorylates Janus kinases (JAK), which then phosphorylate both the canonical recruitment sites for STAT3 and the STAT3 proteins themselves (*p*-STAT3), resulting in the formation of *p*-STAT3 homodimers. The translocation of this homodimer to the nucleus leads to increased expression of Zinc finger E-box-binding homeobox (ZEB), a key transcriptional regulator of partial dedifferentiation and pEMT transition, which is a necessary step enabling the proliferation of pTECs. The autoregulatory production of IL-11 in pTECs perpetuates this process, induced by both IL-11 and TGFβ1. In conjunction with the Wnt-induced β-catenin/TCF pathway and the MKK3/p38 targets of TGFβ1 [40], pTECs are expected to proliferate and differentiate into TECs, facilitating the repopulation of damaged cells. Interestingly, SNAI1, a repressor of this process, is expressed downstream of both IL-11 and TGFβ1. However, it remains inactive due to its continuous translocation to the cytosol for proteolytic cleavage. Canonical IL-11 and TGFβ1 signaling stimulate myofibroblasts for the transient production of ECM, supporting TECs in completing the repair and achieving physiological resolution of kidney damage (Figure 4A).

The pathological setting likely includes the alternative IL-11 signaling, which involves the occupation of the Y759 tyrosine on the gp130 receptor by the SHP2/GRB2/Ras/Raf/MEK pathway, leading to the phosphorylation of ERK (*p*-ERK) (Figure 3). Subsequent translocation of *p*-ERK to the nucleus and phosphorylation of SNAI1 (pSNAI1) repress the Wnt-induced β-catenin/TCF pathway, resulting in the arrest of the cell cycle at the G2/M phase [33,34,38,45,49,52,53,54,55]. The pTEC becomes locked in a dedifferentiated state and a perpetuated state of IL-11 secretion, stimulating ECM production by myofibroblasts (Figure 4B). These interrelated pathways TGFβ1/IL-11/MEK/ERK are termed TIME signaling [52,56]. These events also result in cellular senescence, which triggers immune signaling and inflammation that cause a progressive cycle of kidney damage detailed in an excellent review by Sheng and Zhuang 2020 [38]. 

Detrimental effects of high IL-11 concentrations have also been evidenced in a liver fibrosis model, where injection of mouse recombinant IL-11 into mice induces liver damage characterized by fibrosis, inflammation, and cell death [57]. Excess of IL-11 is thought to be the cause of the cardiorenal syndrome, in which kidney disease causes heart disease and vice versa, as IL-11 contributes to both kidney and cardiac fibrosis [58]. The use of human recombinant IL-11 in human trials for these diseases confirmed these detrimental results and did not move past the first phase, often because IL-11 caused worsening of disease or cardiorenal syndrome [33]. Moreover, IL-11 was found to promote tumor growth and metastasis, and the presence of IL-11 is now used as a biomarker for poor cancer prognosis [59]. In renal disease, the feedforward production of IL-11 in pTECs is held responsible for the transition to CKD (Figure 4B). 

This hypothesis presented in Figure 4B is supported by the tremendous therapeutic potential of IL-11 inhibition demonstrated in preclinical models using mice deficient in either IL-11 or IL-11 receptors [33,34,50]. Indeed, Widjaja et al. (2022) demonstrate that blocking IL-11 successfully prevents kidney disease in IL-11 knockout mice subjected to acute kidney injury (AKI) [34]. Furthermore, anti-IL-11 treatment reverses kidney disease in a mouse model of CKD [34]. A related study by Widjaja et al. (2022) similarly shows that treatment with anti-IL-11 and angiotensin-converting enzyme inhibitors (ACEi) synergistically improved clinical, molecular, and histopathological biomarkers of kidney disease and extended lifespan in a mouse model of Alport syndrome, a genetic condition that affects hearing, vision, and the glomerular basement membrane and can cause CKD [53]. Finally, IL-11 possesses several characteristics that render it a safe therapeutic target compared to IL-6. Notably, IL-11 remains undetected in the serum of healthy individuals due to its predominant regulation of extracellular matrix (ECM) rather than immune responses, favoring alternative ERK activity over JAK/STAT3 activity in chronic kidney disease (CKD) settings [33,49,50,60]. Thus far, data from anti-IL-11 therapy in both mice and humans demonstrate a robust safety profile, with no significant toxic side effects or serious loss of function observed in the long-term administration [49]. Genetic deletion of IL-11R in mice results in dental abnormalities, skull deformities, and infertility, particularly in female mice [61]. Humans with IL-11R mutations have mild craniosynostosis, abnormal tooth eruption, joint laxity, and scoliosis, and, unlike mice, women with the mutation may be fertile [61]. IL-11 knockout mice have even milder deficits compared to wild type: moderately lower body weight, no changes in bone or skull abnormalities or blood composition, and infertility in females as well as reduced fertility in males [61]. Cook (2023) proposed that IL-11 serves as a redundant vestige in mammals, contributing to fibro-inflammatory disease following tissue injury. This suggests that IL-11 evolved from blastema formation, which is driven by IL-11, to regenerate wounded limbs, tails, fins, and even organs in lower vertebrates [32]. Given that mammals cannot form blastema, the upregulation of IL-11 leads to a dysfunctional increase in ECM, resulting in fibrosis, scarring, and inflammation rather than regeneration after organ injury [32]. Unlike IL-6, IL-11 does not appear to have any essential functions in health; therefore, anti-IL-11 treatment is unlikely to induce the immunosuppressant effects seen with anti-IL-6 medications [32,50]. Hence, treatment with pharmacological agents, specific nutrients, or dietary patterns that suppress IL-11 production and secretion holds promise for enhancing resistance against fibro-inflammatory pathogenesis in the kidneys and other diseases, with minimal associated side effects.

## 4. Other Contributors to Kidney Repair and Regeneration

Kidneys possess a degree of repair and regeneration capability following AKI, though this remains a contentious area of research. The primary restoration of kidney function appears to take place in the proximal tubules, and there is a prevailing belief that nephron regeneration is not feasible in mammals, despite being observed in lower vertebrates like zebrafish [62,63,64]. Except for mechanisms discussed in Section 2 and Section 3 about the TGFβWNT/IL11 signaling cascade, the mechanisms of repair are still largely unknown [64], even though stem/progenitor cells found in different parts of the kidney are believed to be recruited from the bone marrow [65]. Extracellular vesicles (EVs) secreted by mesenchymal stem cells are believed to play a role in AKI repair and regeneration, potentially through autocrine/paracrine signaling or by directly regulating TEC proliferation [66,67]. Therapies involving stem cells and EVs derived from mesenchymal stem cells in rodent models and human pilot studies show potential to protect kidneys from injury and inflammation and promote regeneration [64,67].

Other factors associated with successful kidney repair after AKI [64,67], rather than a maladaptive transition to CKD, include the following:(1)Cellular death by apoptosis instead of increased cellular senescence and death by more inflammatory necrosis.(2)Maintenance of macrophage plasticity with a normal ability to polarize between M1, M2, and other phenotypes in response to different signaling environments, typically with a lower ratio of M1:M2 macrophages in the repair phase of AKI.(3)Mild podocyte injury signals parietal epithelial cells or stem cells to participate in the regeneration of podocytes, instead of more severe podocyte injury that signals activated parietal epithelial cells to participate in invasive glomerular hyperplasia [68,69].

Identifying diets that influence the traits supporting kidney regeneration could be regarded as a promising prevention strategy against CKD. 

## 5. Dietary Patterns and Their Impact on Preventing and Managing Kidney Disease

GFR is a critical measure of kidney function, representing the cumulative filtration capacity of all functional nephrons. The average individual with two kidneys possesses approximately 1.8 million functional nephrons. A 50% loss of functional nephrons necessitates the remaining nephrons to double their efficiency to sustain a normal total GFR [8]. A meal high in animal protein, sodium, and inorganic phosphates leads to glomerular hyperfiltration, characterized by increased renal blood flow and GFR [6,70,71]. Over time, glomerular hyperfiltration may lead to nephron hypertrophy, as filtration surface area increases to compensate for increased workload, as well as impaired autoregulation, making the nephron more vulnerable to systemic hypertension and susceptible to insult from local hypertension causing AKI [8,71]. A typical meal in the Western diet, characterized by a significant intake of animal protein and processed foods containing high levels of sodium and phosphate additives, alongside limited fruits and vegetables, represents a common chronic kidney stressor. While a young, healthy individual with over 75% kidney function may handle this dietary load adequately, it could exacerbate kidney damage in an older individual with less than 25% kidney function [70]. 

Only a limited number of clinical trials have investigated the influence of dietary patterns, foods, and nutrients on the risk of developing CKD, as depicted in Appendix A. Most of the studies involve populations with chronic diseases such as CVD or T2DM, or demographic categories such as ethnicity or income, which increase the risk of developing CKD for study participants. Many of these studies are observational or post hoc analyses related to larger studies, such as PREDIMED [72,73] and ARIC [74]. The majority of trials reviewed in Appendix A explore correlations between adherence to a dietary pattern or frequency of nutrient, food, or food group intake and health outcomes, including the risk of eGFR decline or incidental CKD cases. However, a common limitation of these studies is the low observed adherence to the investigated dietary pattern [8,75].

Appendix A presents evidence supporting a negative correlation between the risk of developing renal disease and the consumption of a healthy diet emphasizing fruits and vegetables, whole grains, and plant-based protein. Conversely, it highlights a positive correlation between the risk of renal disease and the consumption of the standard Western diet, characterized by high levels of refined grains, added sugars, animal-based protein, sodium, and saturated fat. Diets such as the Mediterranean diet (MedDiet) with olive oil, MedDiet with nuts, low-fat diet [72], energy-reduced MedDiet [76], and diets rich in fruits and vegetables [77] have been associated with maintaining or improving eGFR. Similarly, the DASH diet, along with a high intake of nuts, legumes, low-fat dairy, magnesium, and calcium [74], as well as HDP [78,79], are linked to a reduced risk of CKD incidence. However, results for the Med diet and DASH diet in preventing CKD are inconsistent, with some studies showing no significant difference in renal outcomes [73,75,76,80,81]. In populations with T1DM or early hypertension (pre- or Stage I HTN), higher adherence to the Healthy Eating Index (HEI) and diets abundant in fruits, vegetables, whole grains, and nuts (FV) is linked to a reduced risk of developing microalbuminuria [75] or decreased urinary albumin excretion rate (AER) in individuals with high-normal baseline AER [82]. On the other hand, diets high in red meat, processed meat, and [74] or low adherence to DASH in the poverty group of a study in an urban community [80] are associated with an increased risk of CKD incidence. One study suggests that diets relatively high in protein and with a high ratio of animal to plant proteins increase the risk of eGFR decline in people with metabolic syndrome and overweight/obesity [76].

At this time, there is no strong evidence to support any particular dietary pattern or macronutrient proportions to prevent CKD. However, a generally healthful diet rich in whole foods, including fruits, vegetables, nuts, legumes, and whole grains, while limiting sodium, added sugars, and saturated fat, is associated with a decreased risk of CKD development, eGFR decline, and/or microalbuminuria in high-risk groups. A possible explanation for this association in the studied populations at risk for CKD is improved management of hypertension and hyperglycemia. Additionally, there is weak evidence suggesting that diets high in protein, especially animal-based protein, red meat, and processed meat, may increase the risk of CKD development and eGFR decline in certain at-risk populations. Further research is needed to study the effects of various dietary patterns as well as protein type and proportion on the risk of developing CKD in healthy populations.

## 6. Dietary Trends Influencing CKD in Clinical Trials

Clinical trials involving baseline CKD, as shown in Appendix A, exhibit a more varied design, including some randomized trials. There is a greater number of studies investigating whether specific dietary patterns, foods, and nutrients might prevent CKD progression, alleviate symptoms and complications of CKD, and reduce the risk of all-cause mortality. The data suggest several mechanisms involved in dietary CKD management:Direct stabilization of kidney function and biomarkers of kidney injury.Decreased metabolic acidosis.Improved phosphorus balance and decreased FGF23 levels.Improved microbiome health, reduced uremic toxins, and alleviated constipation.Reduced inflammation and oxidative stress.Effective weight management, maintenance of lean mass, and nutritional status.Decreased cardiovascular and coronary artery disease risk, as well as reduced dyslipidemia and blood pressure.Lowered all-cause mortality.

Several dietary patterns and nutrients directly support the maintenance of kidney function. Both soy and standard low-protein diets are associated with a decreased rate of CKD progression [83], and a vegetarian very-low-protein diet with supplemental ketoanalogues (SVLPD) is linked to stable eGFR and delayed initiation of renal replacement therapy [84]. Adherence to a healthy dietary pattern (HEI, MedDiet, or DASH) and particularly adherence to MedDiet and increased intake of vegetables and nuts are associated with a decreased risk of CKD progression [85]. The Italian Organic Mediterranean Diet (IMOD) is associated with decreased albuminuria [86]. Soy protein is linked to improved kidney biomarkers or decreased eGFR [87,88]. Finally, treatment of metabolic acidosis by either increasing intake of fruits and vegetables or taking oral sodium bicarbonate is associated with stable eGFR and/or decreased urine indices of kidney injury [89,90,91]. Other diets and foods associated with decreased metabolic acidosis include the New Nordic Renal [92], vegetarian SVLPD [84], and plant-based protein [93]. It is important to note that some studies of diets involving increased fruits, vegetables, and plant-based proteins also assessed the risk of hyperkalemia; however, they generally found no increased risk [93,94]. Nonetheless, one study observed increased urinary excretion of potassium and, therefore, potential for hyperkalemia in patients with low eGFR or other risk factors [90]. Diets associated with weight management and maintenance of lean mass and nutrition status include IMOD [86], vegetarian SVLPD [84], fruit and vegetable treatment of metabolic acidosis [67,90,91], both soy and standard low-protein diets [83], and soy-based protein supplements [95]. They address malnutrition and protein-energy wasting, which are serious concerns in CKD associated with poor prognosis.

CKD often results in abnormal phosphorus balance, which contributes to osteoporosis and heart disease; dysbiosis with increased production of uremic toxins; and increased systemic inflammation and oxidative stress. Diets associated with improved phosphorus homeostasis and decreased FGF23 include IMD and IMOD [86], the New Nordic Renal Diet [92,96], vegetarian SVLPD [84], and plant-based protein [93]. Several diets are linked to improved microbiome, decreased uremic toxins and urinary excretion of urea nitrogen, and reduced constipation: New Nordic Renal Diet [92,96], vegetarian SVLPD [84], diets with high quality based on plant-based dietary index, food groups, fiber intake, and dietary protein-to-fiber ratio [97], added fiber [98], soy low-protein diet [83]; and diets associated with decreased inflammation and oxidative stress include MedDiet [99], soy-based protein supplement [95], and freshly squeezed pomegranate juice [100].

Patients with CKD are at increased risk of CVD, a leading cause of death. Dietary patterns and foods associated with decreased risk of CVD, improved dyslipidemia, lower blood pressure, and/or decreased risk of coronary artery disease include IMD and IMOD [86], adherence to a healthy dietary pattern (HEI), MedDiet, or DASH, and especially adherence to MedDiet and increased intake of vegetables and nuts [85], fruit and vegetable treatment of metabolic acidosis [89,90,91], soy protein [83,87,88,101,102], added fiber [98], freshly squeezed pomegranate juice [100], and both fish oil and olive oil supplements [103].

Finally, a meta-analysis of healthy eating patterns in patients with Stages 2–5 CKD, which included studies with a total of over 15,000 participants, found an association between increased adherence to a healthy dietary pattern and decreased risk of all-cause mortality [104]. Therefore, the evidence does not point to a single beneficial dietary pattern for CKD; namely, MedDiet, DASH, HEI, New Nordic Renal Diet, vegetarian SVLPD, diets high in fruits and vegetables, and soy or other plant-based-protein diets all appear to improve the health of patients with CKD. A wide variety of diets rich in fruits and vegetables, whole grains, fiber, nuts, and legumes; low to moderate in protein; and limited in sodium, added sugar, and saturated fats comprise dietary patterns beneficial for CKD patients, even though the mechanisms underlying these effects remain uncertain.

## 7. Application of Nutrient Enrichment Strategies for Targeting Specific CKD Mechanisms

Regenerative and repair mechanisms, including the inhibition of IL-11 and ERK signaling, systemic and local inflammation, and/or pathways influencing stem cell recruitment, could represent possible mechanisms of the effects of healthy dietary patterns in reducing both CKD progression and the risk of all-cause mortality. The number of studies and interventions discussed below highlighted several phytochemicals, and nutrients, that might target inhibition of IL-11 to decrease renal pEMT and fibrosis include increased dietary intake or supplementation with lutein and other carotenoids, curcumin/turmeric, quercetin, osthole/coumarin, allicin, β-elemene, rosmarinic acid, and omega-3 fatty acids (ω3FA). 

The green and yellow vegetables and fruits, such as leafy greens, herbs, broccoli, peas, green bell peppers, and squash, are relatively rich sources of the oxycarotenoids lutein and zeaxanthin [105,106]. Lutein prevented the overexpression of IL-11 and ERK signaling that occurs in response to angiotensin II infusion and protected mouse hearts from fibrosis, oxidative stress, and pathological cardiac remodeling in a mouse model of cardiac remodeling and heart failure [107]. These pathways are pertinent to CKD and offer a plausible mechanistic framework along with potential biomarkers, warranting further mechanistic elucidation and validation in upcoming clinical investigations. Observational studies in human subjects with CKD have delved into diets supplemented with carotenoids (Table 1). Increased serum lutein is associated with decreased incidence of diabetic kidney disease, increased eGFR, and decreased mortality [108,109,110]. Astaxanthin supplementation did not show a significant difference in arterial stiffness, oxidative stress, or inflammation after renal transplant [111]. Vitamin A supplementation after pyelonephritis in pediatric patients showed improvements in AKI with a decreased rate of permanent kidney damage and decreased renal scar development and injury [112,113,114]. It is noteworthy that elevated plasma retinol levels were linked to a significant decrease in eGFR, indicating a potentially detrimental effect [109]. Indirectly, these studies have suggested that oxycarotenoids, such as lutein and zeaxanthin implicated in IL-11 inhibition, appear to hinder CKD progression, in contrast to vitamin A and astaxanthin, which exert different actions. However, the underlying mechanisms and the involvement of IL-11/ERK axes await further exploration.

Phytochemicals, including quercetin [115,116], curcumin/turmeric [117], and the polyphenol rosmarinic acid found in many herbs like rosemary, thyme, and peppermint, mitigate the overactivation of TGF-β and ERK signaling in diabetes and related diabetic nephropathy and fibrosis [118]. A study examining the impact of nanocurcumin on inflammatory cytokines revealed significant downregulation of IL-11 during the acute inflammatory phase compared to the control nanocarrier in a rat model of spinal cord injury [117]. Curcumin supplementation in human subjects with CKD demonstrates multiple benefits, including the stabilization of eGFR and uremic toxins, reduction in inflammation and oxidative stress, enhancement of antioxidant capacity, improvement of lipid profile, and enhancement of the gut microbiome, as summarized in Table 2 [119,120,121,122,123]. 

One clinical study [124] investigated the effects of isolated 225 mg quercetin on human subjects with Stages 1–5 CKD for 12 weeks (Table 2). The study found no significant difference in kidney function biomarkers (e.g., changes in eGFR), endothelial dysfunction biomarkers, inflammation, oxidative stress, or risk for adverse events between the isoquercetin group and placebo. The low absorption of phytochemicals is often regarded as a significant limitation in supplementation trials. However, after a single dose of 100 mL of concentrated red grape juice containing natural polyphenols and quercetin, it is well absorbed in both healthy patients and clinically stable ESRD patients on hemodialysis [125]. Furthermore, when administered twice daily at 50 mL for 2 weeks, this supplement improves risk factors for cardiovascular disease, including increased antioxidant capacity and an improved lipid profile in ESRD patients undergoing hemodialysis [125]. A twice-daily bioflavonoid supplement, Oxy-Q, containing 480 mg curcumin and 20 mg quercetin, initiated one month after renal transplant surgery in ESRD patients, demonstrated improved early graft function with fewer incidences of acute graft rejection within 6 months and reduced drug-induced tremor [126]. Only one clinical trial has investigated the effects of the Canephron N supplement, which contains rosmarinic acid, centaury herb, lovage root, and rosemary leaves, in human subjects with early diabetic nephropathy. The trial demonstrated a beneficial decrease in albuminuria and oxidative stress; however, it did not show a significant difference in eGFR [127]. Concentrated supplements with phytochemicals, coupled with alternative delivery technologies, may augment their potential efficacy in renoprotective interventions in the future [115,128,129].

The phytochemicals, such as allicin found in garlic, osthole (a coumarin derivative found in the *Umbelliferae* and *Rutaceae* families) [130], and a terpenoid β-elemene found in mint and celery [131], emerge as renoprotective due to their suppression of IL-11 and MAPK/ERK pathways in experimental models [132]. Allicin decreased albuminuria and renal fibrosis, accompanied by reduced expression of TGFβ1 and *p*-ERK1/2 in a rat model of STZ-induced diabetic nephropathy [132]. Osthole was studied in a mouse model of AKI inflicted by surgical ureteral obstruction and in vitro in the human proximal TECs [130]. These researchers found decreased renal fibrosis through inhibition of IL-11/ERK signaling, as well as TGFβ/Smad2/3 signaling. A similar study was performed with β-elemene using the same mouse model of AKI and in vitro rat interstitial fibroblast cells stimulated by TGFβ. Treatment with β-elemene decreased the expression of fibrotic proteins and inhibited SMAD signaling [131]. 

**Table 2 nutrients-16-01342-t002:** Clinical trials involving phytochemicals.

Nutrient	Outcome	Duration	Baseline CKD Stage and Non-CKD Conditions/Criteria	Source
Canephron N (centaury herb, lovage root, rosemary leaves) supplement/enalapril (n = 36). Enalapril only (C) (n = 23).	Canephron N: ↓ albuminuria, NSD in eGFR, ↓ oxidative stress, no adverse effects.	6 mo.	Stage 1–2. T2DM for at least 6 months with microalbuminuria (UA > 30 mg/day or ≥ 3 mg/dL or UACR < 2.26 mg/mmol).	[127]
Curcumin (Merida) 500 mg BID supplement in CKD (n = 24 at T0, n = 21 at T1, n = 11 at T2). Age/sex matched healthy control for normal microbiome (n = 20 at T0).	Stable eGFR and uremic toxins, improved gut microbiome, ↓ inflammation biomarkers, and no adverse effects.	3 mo (T1), 6 mo (T2).	Stage 3a, 3b, and 4. N/A.	[119]
Curcumin 320 mg/day supplement (n = 28 diabetic, n = 24 nondiabetic). Oral placebo/day (C) (n = 23 diabetic, n = 26 nondiabetic).	↓ lipid peroxidation in nondiabetic CKD, ↑ antioxidant capacity in diabetic CKD, but NSD in proteinuria, eGFR, lipid profile, or markers of oxidative stress and inflammation.	8 weeks.	Stage 3–4. Curcumin and placebo groups subdivided into nondiabetic or diabetic CKD with proteinuria (UP ≥ 1 g/day).	[120]
Orange–carrot juice/curcumin (n = 14). Orange–carrot juice only (C) (n = 14).	↓ inflammation biomarkers, ↓ NF-kB mRNA expression. NSD other markers and no adverse effects.	3 mo.	ESRD, HD ≥ 6 months.	[121]
Orange–carrot juice/curcumin (n = 14). Orange–carrot juice only (C) (n = 14).	↓ uremic toxin, p-cresyl sulfate NSD in uremic toxins, indoxyl sulfate or indol acetic acid	3 mo.	ESRD, HD ≥ 6 months.	[133]
Curcumin 1 g/day supplement (n = 20). Oral placebo/day (C) (n = 23).	NSD in markers of oxidative stress and inflammation.	12 weeks.	ESRD, HD ≥ 3 months.	[122]
Curcumin 500 mg BID supplement (n = 35). Oral placebo/day (C) (n = 36).	NSD in ↓ inflammatory markers.	12 weeks.	ESRD, HD ≥ 3 months, clinically stable.	[123]
Curcumin 500 mg/resveratrol 500 mg daily supplement (n = 20). Oral placebo daily (C) (n = 20).	↑ bone density and muscle mass, ↓ iron overload.	12 weeks.	ESRD, HD.	[134]
Low dose (daily) 480 mg curcumin/20 mg quercetin (Oxy-Q) supplement (n = 14). High dose (BID) 480 mg curcumin/20 mg quercetin (Oxy-Q) supplement (n = 14). Oral placebo BID (C) (n = 15).	Improved early graft function. Improved early graft function, ↓ incidences of acute graft rejection within 6 months, ↓ drug-induced tremor.	6 mo.	ESRD, renal transplant.Supplement started 1 month after renal transplant surgery.	[126]
Quercetin in concentrated RGJ (n = 26 HD, n = 15 healthy). Control (HD usual care) (n = 12).	Improved CVD risk factors and lipid profile, ↑ antioxidant capacity, ↓ inflammation biomarkers. Note: Healthy quercetin subgroup confirmed good bioavailability from RGJ.	14 day supplement, 6 mo follow-up.	ESRD, HD ≥ 3 months, clinically stable. Quercetin group subdivided into HD and healthy groups.	[125]
Isoquercetin 225 mg/sodium nitrite 40 mg daily (n = 35). Oral placebo/sodium nitrite 40 mg daily (n = 35).	Isoquercetin/sodium nitrate: NSD in endothelial dysfunction biomarkers, kidney function, or risk for adverse events.	12 weeks.	Stage 1–5. Stage 5 was predialysis (eGFR of 12–105 mL/min).	[124]

↓ significantly decreased and ↑ increased compared to control group.

## 8. The Role of Omega-3 Fatty Acids in Counteracting CKD Progression

Prevention of CKD in high-risk individuals, managing CKD symptoms, and maintaining kidney function through ω3FA have been understudied or have shown inconsistent results in clinical trials. Supplementation studies utilizing EPA/DHA from fish (Appendix A) provide evidence of a slower annual decline in eGFR among individuals after myocardial infarction with CKD compared to patients with early or no CKD [135]. Meta-analysis supports a very low certainty regarding the reduced risk of CKD progression to ESRD and shows no effect on the risk of all-cause mortality or transplant rejection in ESRD patients supplemented with ω3FA from fish [136,137] (Table 3). Seafood ω3FA supplementation is associated with a decreased risk of developing CKD and a reduced annual decline in eGFR in individuals without CKD, but supplements with α-linolenic acid (αLA) or ω3FA from plants have shown no significant difference in preventing CKD [135,138]. Disregarding the fish or plant origin, ω3FA did not influence incident CKD or rapid eGFR decline [135]. However, a relatively high intake of nuts containing ω3FA was associated with a decreased incidence of CKD in one observational study of individuals without CKD in selected communities [74]. Moreover, αLA from a seed mixture was found to decrease uremic pruritis in ESRD patients [139]. Collectively, the results of these studies suggest that ω3FA may interact with other compounds to influence systemic processes that affect CKD.

Many findings suggest that ω3FA primarily influence lipid metabolism, reducing the risk of CVD and thereby decreasing CKD pathogenesis. Table 2 and Table 3 demonstrates that supplementation with ω3FA from fish is linked to increased fibrinolytic potential, decreased inflammatory biomarkers, improved cardiometabolic parameters (such as decreased plasma levels of triacylglycerols, total cholesterol, oxidative stress, and phosphorus), and low-certainty reduction in the risk of CVD death in ESRD patients on hemodialysis [103,136,137,140,141]. However, in CKD patients, supplementation with ω3FA from fish yields variable results in lowering blood pressure, LDL, and some inflammatory markers, with no significant difference observed in HDL [136,137,140,141]. Plant-based αLA has also been associated with a significant decrease in inflammatory markers contributing to CVD risk reduction [139,142], but it has not been shown to make a significant difference in the lipid profile of CKD patients [142]. These studies argue for higher efficacy of supplement ω3FA from fish than αLA in reducing cumulative CVD risks in CKD patients.

Mechanistic studies in mouse models and human cancer revealed that IL-11 production and phosphorylation of ERK are candidate targets of ω3FA, which could contribute to the reduction in inflammatory and CKD progression observed in clinical studies (Table 3). Hagiwara et al. found that EPA inhibits phosphorylation of ERK1/2 in the glomeruli, reduces mesangial ECM and tubulointerstitial fibrosis, and decreases hypertriglyceridemia and other CVD risk factors in a mouse model of diabetic nephropathy [143]. DHA supplementation reduced inflammation and renal injury, coinciding with decreased ERK phosphorylation in the kidney, in a sepsis mouse model of AKI [144]. This mechanism was also at work in other tissues. Mitigation of acetaminophen toxicity by ω3FA depends on the inhibition of IL-11 expression and phosphorylation of ERK1/2, thus blocking hepatic injury [145]. In human breast cancer cells, DHA inhibits gremlin-1 (GREM1), thereby preventing the activation of the ERK/*N*-cadherin/Slug axis and partial EMT [146]. GREM1 is also a key regulator in kidney fibrosis [147]; however, the mechanism of interaction between DHA and GREM1 in CKD remains unknown. The discrepancy between the limited efficacy of ω3FA in improving CKD progression in vivo and their effective regulation of IL-11 in vitro raises questions about the mechanisms of ω3FA uptake in renal tissues and their reliance on dietary sources and inflammation. 

## 9. Vitamin D Deficiency and CKD Progression

Vitamin D metabolism is intricately linked to kidney function, as it involves the conversion of vitamin D to the hormone calcitriol, which plays a pivotal role in regulating calcium and phosphate homeostasis (reviewed in [148]). The kidneys are responsible for the synthesis, and catabolism of this hormone. Consequently, both deficiency and excess of vitamin D can profoundly impact repair mechanisms and/or exacerbate the progression of CKD, depending on vitamin D levels [149,150]. Therefore, maintaining optimal levels of vitamin D in patients is paramount for the effective management of CKD. Vitamin D deficiency is considered to contribute to age-related diseases, including CKD, although the underlying mechanisms remain largely unexplored [151]. Serum levels of vitamin D begin to decrease as early as stage 2 CKD, with 97% of ESRD patients on hemodialysis experiencing vitamin D deficiency [152]. Vitamin D deficiency is associated with the stage of CKD and promotes CKD progression, leading to the development of renal osteodystrophy and an elevated risk for all-cause mortality [152,153]. Moreover, reduced levels of serum vitamin D are linked to increased oxidative stress and exacerbated secondary hyperparathyroidism [152]; hence, the supplementation of vitamin D in different forms was anticipated to be advantageous for CKD management with minimal associated risks. However, a recent meta-analysis of randomized controlled trials reports contradicting results ranging from harmful side effects of supplementation, like hypercalcemia, and clinical benefits, including statistically significant reductions in bone fractures, improved cardiovascular outcomes, and decreased risk for all-cause mortality with ordinary doses of vitamin D supplements [150,154,155] to the absence of any effects on mortality [156]. Vitamin D supplementation is no longer routinely recommended for the general population or CKD patients, whereas clinical attention is given to addressing vitamin D deficiency [157]. Considering the limitations of the trials included in the meta-analysis [157,158], there is an emerging focus on optimizing the therapeutic dose, form, and utilization of vitamin D supplementation to benefit CKD patients.

Identifying a surrogate marker for “optimal” vitamin D levels necessitates understanding the mechanisms that disrupt renal metabolism in states of vitamin D deficiency. Recent studies have uncovered a paradoxical finding in CKD, where vitamin D deficiency is accompanied by overexpression of fibroblast growth factor-23 (FGF-23), despite FGF23 levels being regulated by the calcitriol-activated vitamin D receptor (VDR) [159]. Vitamin D deficiency appears to be secondary to FGF-23 overproduction. Correction of FGF-23 overproduction by C-type natriuretic peptide (CNP) normalizes vitamin D levels, *p*-ERK signaling, and renal dysfunction in uremic rats. Consequently, secondary hyperparathyroidism and renal osteodystrophy in uremic rats were also improved [159]. The association of vitamin D deficiency and TGFβ1/IL-11/*p*-MEK/*p*-ERK-dependent fibrosis was also observed in a genetic mouse model of vitamin D deficiency mimicking some effects of FGF-23 overproduction [160]. This study demonstrated that IL-11-dependent fibrosis could be alleviated by sirtuin 1 (SIRT1), which controls the transcription of IL-11 by deacetylating its promoter region, activated by SMAD2 [160]. These data suggest that vitamin D deficiency in renal disease should be interpreted and treated in conjunction with endocrine abnormalities to enhance the efficacy of dietary vitamin D interventions involving this hormonal precursor. The dosing of dietary vitamin D is critical, as both oversupplementation [161] and deficiency of vitamin D [162] are associated with dysbiosis and inflammation in the gastrointestinal tract, impacting the progression of CKD.

## 10. The Microbiome and Dietary Influences on Mechanisms Preventing CKD Progression

Gut dysbiosis is associated with CKD [163], and it has harmful effects that contribute to CKD progression, like uremic toxins, reduced short-chain fatty acid (SCFA) production, inflammation, and oxidative stress [164]. The kidney–gut axis is bidirectional, with uremia causing gut dysbiosis and gut dysbiosis producing uremic toxins that contribute to uremia [164]. Improvement of microbiome diversity to slow CKD progression was achieved through diets such as the New Nordic Renal Diet [92,96] and vegetarian SVLPD [84,97], enriched in high-quality proteins, prebiotics, probiotics, and phytochemicals, as well as interventions such as fecal microbiota transplantation [164]. Indeed, fecal microbiota transplant from an animal with CKD to a healthy rodent induces the production of uremic toxins and interstitial fibrosis. Conversely, fecal microbiota from a healthy animal to an animal with CKD reduces the production of uremic toxins and decreases tubulointerstitial injury [164]. 

It is plausible that IL-11 is involved in this gut–kidney axis because it plays a role in communicating between gut dysbiosis and tumor metastasis [59], as well as gut dysbiosis and liver failure [165]. Zhu et al. (2020) found that broad-spectrum antibiotics, which deplete fecal microbiota, promote tumor metastasis [59]. Conversely, either fecal microbiota transplant from healthy mice into germ-free mice or inoculation of healthy Bifidobacterium into germ-free mice inhibits tumor metastasis. This inhibition of tumor metastasis, achieved by improving the gut microbiome, occurs through EMT signaling pathways, including the downregulation of IL-11 in tumors, thereby reducing cancer progression [59,165]. Microbiomes generate numerous bioactive metabolites, dependent on the diet composition, and could potentially provide a source for the discovery of pharmacological inhibitors of IL-11. Lactobacillus reuteri DSM 17938 reduces d-galactosamine-induced liver injury, in part by decreasing the transcription of inflammatory cytokines IL-11 and IL-6 [165]. More studies need to be conducted to elucidate the effects of these probiotics on renal IL-11 production and CKD outcomes.

## 11. Conclusions and Perspectives

The remarkable progress in elucidating dietary patterns (Appendix A), specific nutrients, and supplements that could improve CKD progression has been significant (Table 1, Table 2, Table 3 and Appendix A). However, the major challenge of regenerating and repairing kidneys in CKD using nutritional approaches remains an enigma. Recent breakthrough discoveries highlighting the critical role of IL-11 overproduction in driving PMT of TECs and fibrosis (Figure 2, Figure 3 and Figure 4) offer an opportunity to explore specific nutrients targeting these pathways. Candidate nutrients discussed in this review primarily include lutein/zeaxanthin and phytochemicals like allicin, osthole, β-elemene, quercetin, curcumin, and rosmarinic acid, which primarily mitigate *p*-ERK/SNAI1 signaling axes exacerbating IL-11 production. In contrast, ω3FA supplements appear to provide indirect benefits for CKD patients, and addressing vitamin D deficiency should be performed in conjunction with endocrine factors, such as the overproduction of FGF23. However, systematic studies are required to pinpoint the mechanisms responsible for the transition of IL-11 from canonical to alternative pathways, as well as to elucidate the dependency of regenerative pathways on specific nutrients.

Given the low absorption of phytochemicals, it is plausible that the complex composition of these molecules, when used at low concentrations, provides more benefits than single-molecule supplementations. Future developments in improved renal dietary patterns may consider substantial additions of herbs containing various phytochemicals at low concentrations and presenting prebiotics counteracting dysbiosis in CKD patients. The direct suppression of IL-11 by SIRT1 necessitates testing additional phytochemicals, for example, resveratrol and ketone bodies [166,167,168], in regulating IL-11 via SIRT1 activation and/or other mechanisms [169,170] implicated in kidney regeneration (Appendix A). Advancements in comprehending the nutritional mechanisms governing IL-11 overproduction and other transitional factors, pivotal in the shift from physiological regeneration to maladaptive repair, hold promise for a rationalized formulation of renal regenerative diets. This objective, though ambitious, stands within foreseeable reach in the near future.

## Figures and Tables

**Figure 1 nutrients-16-01342-f001:**
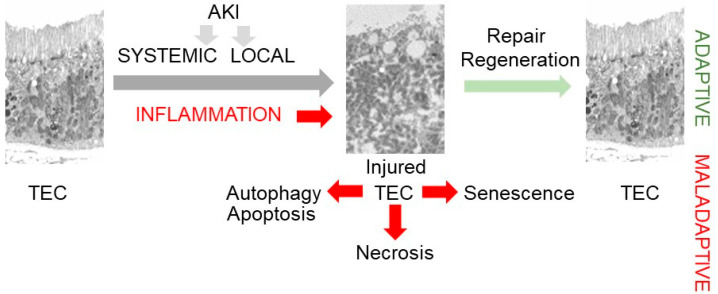
Schematics of adaptive (green) and maladaptive responses (red) in renal tubule epithelial cells (TECs) following systemic or local injury. TEC injury commonly results from hypoxic stress, which can be caused by a systemic decrease in cardiac output or local obstruction of circulatory renal filtration due to strictures, thrombus, tumors, scarring, or other factors. Adaptive and maladaptive response of injured TECs involves fibrosis and inflammation. The adaptive response includes the repair of injured TECs or the apoptosis and autophagy of severely damaged TECs. The maladaptive immune response is triggered by deficient energy production in hypoxic environments, resulting in the formation of senescent cells and senescence-associated secretory phenotype or transition of injured TECs into proinflammatory mesenchymal VCAM^+^ phenotype. These cells release proinflammatory TGFβ, IL-11, Wnt, PDGF, and Snail 1, stimulating excessive fibrosis by myofibroblasts and recruitment of immune cells leading to chronic kidney inflammation.

**Figure 2 nutrients-16-01342-f002:**
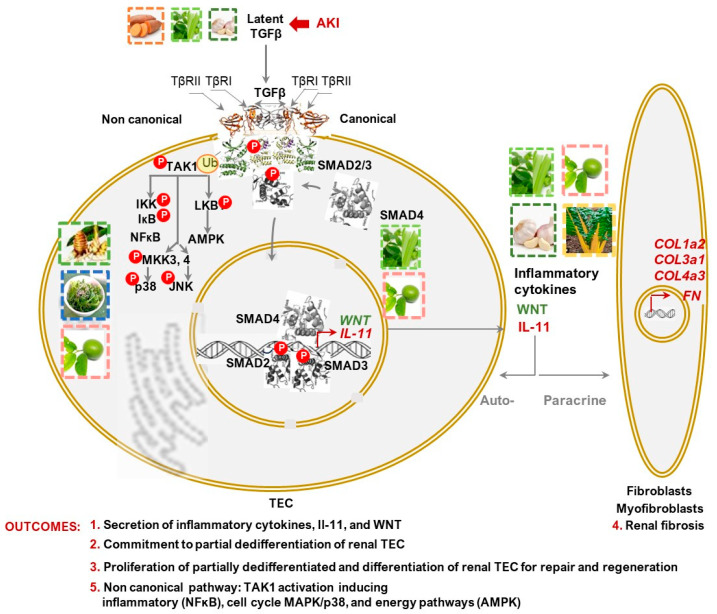
Schematic representation of the TGFβ pathway in tubular epithelial cells (TECs) during systemic or local acute kidney injury (AKI). Injury triggers the proteolytic cleavage of latent TGFβ into its active, free form. The oligomeric complex is formed by TGFβ receptors (TβRI and TβRII). Upon activation via binding with TGFβ, this complex recruits and phosphorylates SMAD2/3 and SMAD4 transcription factors. This canonical pathway of activation leads to their translocation to the nucleus, where they regulate the expression of target genes such as IL-11 and Wnt [43], and transcriptional repressor SNAI1 [48] (Outcomes 1–3). TGFβ and IL-11 collaboratively enhance the levels of ECM for repair (Outcome 4). In a noncanonical pathway of TGFβ activation, phosphorylated TAK1 triggers inflammatory, cell cycle, differentiation, and energy pathways (Outcome 5). The TGFβ pathway occurs in multiple cell types, including stromal fibroblasts, myofibroblasts, and immune cells, in addition to TECs. Nutrients sources encompass phytochemicals implicated in regulating these processes and are elaborated upon in Section 7 and Section 8, and Appendix A. Dashed lines represent the hypothetical pathways investigated in animal models.

**Figure 3 nutrients-16-01342-f003:**
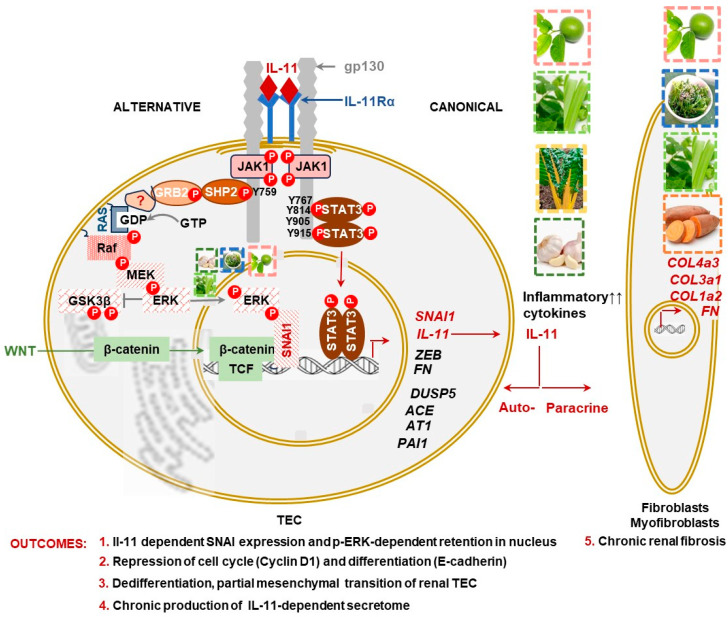
Schematic representation of the IL-11 pathway in TECs during inflammation: The canonical pathway: Upon activation by IL-11 binding, gp130 and IL-11α receptors complex activates JAK1, which phosphorylates gp130 for the recruitment and phosphorylation of STAT3 transcription factors, mediating the expression of IL-11 and other inflammatory cytokines, transcriptional proteins SNAI1 and ZEB, responsible for the partial dedifferentiation and partial mesenchymal transition of TECs (pTECs), along with other genes regulating coagulation, osmotic adaptation, and ECM (Outcomes 1–4). This physiological IL-11 activation supports autocrine and paracrine pathways (Figure 4). Physiological IL-11 stimulation of myofibroblasts restores ECM, and chronic ECM production by pathological levels of IL-11 leads to renal fibrosis (Outcome 5). The IL-11-dependent alternative pathway involves different phosphorylation sites recruiting SHP2 for the activation of the GRB2/RAS/RAF/MEK cascade, culminating in ERK phosphorylation (p-ERK) and its translocation to the nucleus. Phosphorylation of SNAI1 by p-ERK prevents its translocation from the nucleus to the cytosol, where it is degraded by GSK3β. In the nucleus, p-SNAI1 binds with the β-catenin/TCF complex, preventing the expression of genes leading to differentiation, recovery, and repair of TEC. In this scenario, IL-11 overrides TGFβ repair mechanisms locking pTEC in a partially differentiated state while promoting feed-forward overproduction of IL-11, inflammation, and fibrosis in myofibroblasts. Nutrient sources/phytochemicals regulating these processes are described in Section 7 and Section 8, and Appendix A (Dashed lines, hypothetical pathways investigated in animals).

**Figure 4 nutrients-16-01342-f004:**
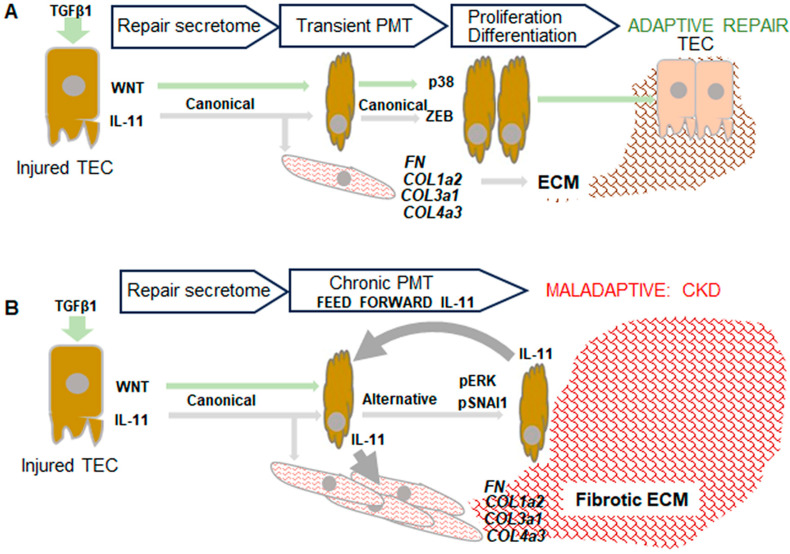
Hypothetical mechanism underlying dissimilar roles of TGFβ1 and IL-11 in adaptive (**A**) and maladaptive (**B**) TEC injury response. Activation of TGFβ1 initiates a cascade of responses aimed at regenerating TECs, which includes the secretion of IL-11 and WNT as part of the repair secretome (Figure 2). Initial IL-11 secretion, in conjunction with TGFβ1, leads to partial mesenchymal transition (PMT); however, WNT-dependent cell cycle stimulation promotes the proliferation of pTECs and their differentiation. Thereby, the TEC layer is repaired and regenerated on the newly produced ECM. The pathological feedforward loop of IL-11 production emerges through its autocrine and paracrine secretion by pTECs arrested in a dedifferentiated state, in which the WNT response is repressed (Figure 3). These pTECs secrete excessive IL-11 levels, stimulating chronic ECM production by myofibroblasts, culminating in renal fibrosis and CKD.

**Table 1 nutrients-16-01342-t001:** Clinical trials involving supplementation with carotenoids.

Nutrient	Outcome	Duration	Baseline CKD Stage and Non-CKD Conditions/Criteria	Source
Astaxanthin (Xanthin) 12 mg/day supplement, carotenoid (n = 32). Oral placebo (n = 26).	NSD in arterial stiffness, oxidative stress, or inflammation after renal transplant.	1 year.	ESRD, renal transplant.	[111]
IM Vitamin A/antibiotics (n = 25). IM Placebo/antibiotics (C) (n = 25).	↓ Rate of permanent kidney damage after pyelonephritis.	3 mo repeat scan.	AKI after pyelonephritis, ages 2–144 months (median 24 months).	[112]
Oral vitamin A/antibiotics (n = 15). Oral vitamin E/antibiotics (n = 18). Antibiotics only (C) (n = 21).	↓ Renal scar development after pyelonephritis. ↓ Renal scar development after pyelonephritis.	≥6 mo (repeat scan).	AKI after pyelonephritis, ages 1 mo to 10 years.	[113]
Oral vitamin A/antibiotics (n = 36). Oral placebo/antibiotics (C) (n = 38).	↓ UTI clinical signs, renal injury, and scarring after pyelonephritis.	6 mo repeat scan.	AKI after pyelonephritis, girls ages 2–12 years.	[114]
Lutein, serum, Q4 * and T4 ** (n = 90). * Q4 indicates highest quartile of all participants ** T4 indicates highest quartile of participants with T2DM.	↓ Incidence T2DM and DKD.	Nontemporal data analysis.	T2DM with CKD (UACR ≥ 30 mg/g) (n = 30). T2DM without CKD (n = 30). Age-matched healthy controls (n = 30).	[108]
↑ Lutein (and zeaxanthin), serum (n = 570). ↑ Retinol, serum (n = 570). ↑ Other antioxidants, serum (n = 570).	↑ eGFR. ↓ eGFR. NSD in sensitivity analysis. NSD.	Nontemporal data analysis.	Stage G3b mean. Alzheimer’s disease (n = 253). Cognitively normal (n = 317).	[109]
Total carotenoid intake, Q4 * (n = 1523 out of 6095, 25%). α-carotene, β-cryptoxanthin, lycopene, and lutein/zeaxanthin, serum, Q4 *. * Q4 indicates highest quartile of intake or serum level.	↓ Mortality after adjusting for confounders. ↓ Mortality.	Mean follow-up time of 8.1 year (NHANES data 2001–2015, endpoint = 31 December 2015 or death.	Stages 1–5. Not pregnant.	[110]

↓ significantly decreased and ↑ increased compared to control group.

**Table 3 nutrients-16-01342-t003:** Clinical trials involving omega 3 fatty acids.

Nutrient	Outcome	Duration	Baseline CKD Stage and Non-CKD Conditions/Criteria	Source
Dietary milled sesame/pumpkin/flax seed mixture, n-6 and ALA n-3 FA (n = 30).	Improved fatty acid profile, ↓ uremic pruritis, IL-6 levels, TNFα levels, and BP.	12 weeks.	ESRD, HD ≥ 6 months, clinically stable. Intervention followed cross-sectional evaluation used as baseline.	[139]
Meta-analysis of ω3FA (n = 4129 across 60 studies).	May have ↓ risk of CVD death in HD patients (low certainty). May ↓ risk of CKD progression to ESRD (very low certainty). NSD risk of all-cause mortality or transplant rejection.	6 mo follow-up.	Stage 1–5. ESRD included HD, PD, transplant.	[136]
Meta-analysis of ω3FA (mostly EPA/DHA, 1 EPA only) (n = 708 across 13 studies).	Improved cardio-metabolic parameters, including ↓ total cholesterol, TG, and oxidative stress. NSD in BP, LDL, or HDL.	8–192 weeks.	Stage 1–5. ESRD included HD, PD.	[140]
Meta-analysis of ω3FA (EPA/DHA capsules) (n = 371 across 8 studies).	↓ CVD risk, ↓ inflammatory markers, CRP and high-sensitivity CRP (hs-CRP). NSD in albumin, TNFα or IL-6.	2.5–6 mo.	ESRD, HD.	[141]
Meta-analysis of ω3FA (n = NI across 49 studies).	↓ CVD risk, TG, LDL, CRP, TNFα and serum phosphorous. Mild GI adverse effects. NSD in albumin, total cholesterol, or all-cause mortality.	1–6 mo.	ESRD, HD, and PD.	[137]
Meta-analysis of plant-based ALA ω3FA (n = 1145 across 19 studies).	↓ CVD risk, inflammatory marker, and CRP. NSD in lipid profile.	4–48 weeks.	Stage 1–5. ESRD included HD, PD.	[142]

↓ significantly decreased and ↑ increased compared to control group.

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
