# Peer review of "Chronic Kidney Disease Diets for Kidney Failure Prevention: Insights from the IL-11 Paradigm"

_nutrients, 2024, doi:10.3390/nu16091342_

Round 1

Reviewer 1 Report

Comments and Suggestions for Authors

Comments on the "Chronic Kidney Disease: New Mechanisms and Nutritional  Strategies to Prevent Kidney Failure".

The authors discussed the physiological resolution of kidney injury by transforming growth factor beta 1 (TGFβ1) and interleukin-11 (IL-11) and the efficacy of plant-dominant diet in CKD.

There are several comments:

1.      The reference 1 is not clear. Website or publication of NIDDK?

2.      The introduction seems too long. For example, Paragraph 4 argued the definition of CKD in the elderly, which is important but still controversial.

3.      The authors possessed similar idea as Prof Cook that IL-11 is redundant. We need more data to confirm that a cytokine is only bad. For example, mechanical stress (exercise) could induce IL-11 which could be a bone-fat communication.

4.      AKI model. Earlier literatures demonstrated anti-apoptosis effect of IL-11 during ischemia reperfusion injury. The model that Prof Cook used is a toxin (FA) induced AKI model. We need more data to confirm the detrimental effect of IL-11 in AKI.

5.      Table 2. The plant-based diet is beneficial, but there is no evidence that these diets decrease IL-11. It seems too far to connect diet pattern with IL-11, before we know which kind of nutrients could affect IL-11. MedDiet and other strategies are not new.

6.      Table 3. The authors provide very few evidences of nutrients that decrease IL-11 in animals and almost none in human. However, the mechanism is not clear. Direct or indirect effect?.

7.      Table 2 and 3 are large. It seems better to divide them by diet method and nutrients.

8.      The authors ambitiously proposed a renal regenerative diet. However, it is still a hypothesis.

9.      The title should be clearer and include “IL-11”. This mechanism is new but the diet strategies are old. Because the nutrient that could affect IL-11 is not clear, there is no new strategy to recommend now.

10.  Focus on potential readers. This review could provide new idea for researchers. But it does not provide new strategy for clinical practicers.

In conclusion, the authors made a good review of IL-11 and suggested a renal regenerative diet. However, the evidences are too weak to change clinical diet pattern. We suggest focusing on evidences and future researches rather than strategies.

Author Response

  1. The reference 1 is not clear. Website or publication of NIDDK?

This is the USRDS Annual Data Report, which is web-based (https://usrds-adr.niddk.nih.gov/2022). We have corrected the citation format accordingly. Thank you for bringing this issue to our attention.

  1. The introduction seems too long. For example, Paragraph 4 argued the definition of CKD in the elderly, which is important but still controversial.

In response to this Reviewer’s suggestion, we have significantly shortened the Introduction (see marked version). We now proceed to present the definition of CKD, age-related controversies surrounding CKD, and details regarding established diets recommended for CKD patients in separate boxes (Box 1-3).

  1. The authors possessed similar idea as Prof Cook that IL-11 is redundant. We need more data to confirm that a cytokine is only bad. For example, mechanical stress (exercise) could induce IL-11 which could be a bone-fat communication.’

We are inclined to support your view rather than that of Dr. Cook. The evidence suggests that IL-11 could play a role in adaptive regeneration (Figure 4A), while an excess of this cytokine in the absence of other differentiation cues (such as WNT) could promote a maladaptive response (Figure 4B).

In Figure 4, we summarize both the favorable physiological role of IL-11 (Fig.4A) and its unfavorable pathological role in kidney injury scenarios involving mesenchymal transition (Fig.4B). Figure 4A depicts the physiological scenario, where transiently produced IL-11 by pEMT cells supports repair processes in conjunction with TGFb and Wnt pathways (described in Fig.2 and Fig. 3). Induction of the cell cycle in pEMT cells and their subsequent differentiation contribute to physiological repair.

In Figure 4B, we illustrate that pERK phosphorylation and its translocation to the nucleus lead to SNAI1 phosphorylation, which blocks the WNT pathway. This event maintains pEMT cells in a dedifferentiated state characterized by increased IL-11 production, thereby inducing a feed-forward loop of this cytokine.

The likely cause of pERK phosphorylation is detailed in Figure 3. Initial IL-11 signaling activates the canonical STAT pathway. However, IL-11 signaling can also initiate an alternative pathway involving Y759 phosphorylation on gp130 (Fig. 3). The precise mechanism for non-canonical pathway activation is not fully understood; however, it is associated with chronic repetitive AKI. This condition may exhaust or saturate the canonical phosphorylation sites, leading to occupancy of the Y759 phosphorylation site and activation of the alternative pathway.

  1. AKI model. Earlier literatures demonstrated anti-apoptosis effect of IL-11 during ischemia reperfusion injury. The model that Prof Cook used is a toxin (FA) induced AKI model. We need more data to confirm the detrimental effect of IL-11 in AKI.

Thank you for raising this point. We are inclined to support your view rather than that of Dr. Cook. The evidence suggests that IL-11 could play a role in adaptive regeneration, while an excess of this cytokine in the absence of other differentiation cues (such as WNT) could promote a maladaptive response. In this review, we delineate both IL-11's role in the physiological beneficial resolution of injury and its maladaptive role, as we described in response to your previous question (Question 3). Although the transition to a non-canonical maladaptive response to IL-11 is not fully elucidated, the use of a toxin (FA) by Prof. Cook could represent chronic exposure leading to pERK phosphorylation. In contrast, ischemia-reperfusion injury could represent an injury model utilizing an adaptive canonical IL-11 response.

We added to our Introduction  ( last paragraph) that we discuss both pathological and physiological aspects of IL-11 signaling.

We also added in the last section : ‘However, systematic studies are required to pinpoint the mechanisms responsible for the transition of IL-11 from canonical to alternative pathways, as well as to elucidate the de-pendency of regenerative pathways on specific nutrients’.

  1. Table 2. The plant-based diet is beneficial, but there is no evidence that these diets decrease IL-11. It seems too far to connect diet pattern with IL-11, before we know which kind of nutrients could affect IL-11. MedDiet and other strategies are not new.

We acknowledge that MedDiets and other dietary pattern strategies are not novel. Given that the description of these clinical trials offers a useful summary for clinical researchers aiming to improve study design, we have relocated it to supplemental data (Table S1, S2).

  1. Table 3. The authors provide very few evidences of nutrients that decrease IL-11 in animals and almost none in human. However, the mechanism is not clear. Direct or indirect effect?.

We appreciate your comment, which highlights the limited understanding of dietary factors influencing IL-11 expression and secretion, as well as the absence of mechanisms for their effects. The objective of this review is to present the available information and raise awareness among researchers about this pathway, thereby paving the way for future investigations into its underlying mechanisms.

  1. Table 2 and 3 are large. It seems better to divide them by diet method and nutrients.

We divided table 3 by nutrients and move Table 2 to Supplementary materials as less novel.

  1. The authors ambitiously proposed a renal regenerative diet. However, it is still a hypothesis.

We rephrase these statements to assert that identifying the mechanisms for the regeneration of CKD is a global goal for the future. Understanding molecular mechanisms may provide the rationale for biomarkers that could be used to identify specific nutrient compositions promoting the regenerative process.

We have rephrased the last paragraph to emphasize that advancements in comprehending the nutritional mechanisms governing IL-11 overproduction and other transitional factors, pivotal in the shift from physiological regeneration to maladaptive repair, hold promise for a rationalized formulation of renal regenerative diets. This objective, though ambitious, stands within foreseeable reach in the near future.

  1. The title should be clearer and include “IL-11”. This mechanism is new but the diet strategies are old. Because the nutrient that could affect IL-11 is not clear, there is no new strategy to recommend now.

First, in response to this reviewer suggestions we replaced our old title ‘Chronic Kidney Disease: New Mechanisms and Nutritional Strategies to Prevent Kidney Failure’

With the new title: ‘CKD Diets for Kidney Failure Prevention: Insights from the IL-11 Paradigm’

Secondly, we have omitted the term "strategies" from the last paragraph of the Introduction, which outlines the scope of our review.

  1. Focus on potential readers. This review could provide new idea for researchers. But it does not provide new strategy for clinical practicers.

Thank you for this advice. Due to the limited research conducted on identifying the link between IL-11 production and nutrients, we adopted a cautious approach to consider the current renal dietary pattern recommendations, which include nutrients implicated in the regenerative processes. The revised figures now also incorporate information about nutrients regulating regenerative processes.

Reviewer 2 Report

Comments and Suggestions for Authors

In general this manuscript is an extensive review on novel mechanisms and nutritional strategies for the management of CKD. The main problem of the manuscript, although it cointains novel end interesting information,  it needs more efford in order to have make more clrea the connection between diet and the mechanisms of kidney repair and regeneration. 

1. The review has not followed the guidelines of scoping or systematic reviews. Therefore it is of limited value.

2. No methodology of searching is presented and no evaluation of the quality of the selected stusied for the review is presented.

3. The manuscript is so long that it looks as if two different reviews are mixed without a clear connection. Moreover, the paragraph on the enrichment is also a third thematic area, important but making the whole document exhaustive. 

4, Tables heading are too extensive and should be shortened. The relative info could be incorporated in the text. 

5. The term elderly should be replaced by older adults in the text. 

6. Figure 1 needs improvement. 

7. I would see this manuscript more useful if the nutrition was more focused on the first part and the 1st part on kidney repair would me more focused on nutrition related aspects. Therefore I would suggest to modify all the figures and the text of 1st part providing more information on how diet can alter the mentioned pathways. 

Comments on the Quality of English Language

Enlgish language use is fine and only the tems older adults should be altered. 

Author Response

Reviewer 2

In conclusion, the authors made a good review of IL-11 and suggested a renal regenerative diet. However, the evidences are too weak to change clinical diet pattern. We suggest focusing on evidences and future researches rather than strategies.

In agreement with this reviewer suggestions we deemphasized ‘strategies’, and reported the findings.

First, we replaced our old title ‘ Chronic Kidney Disease: New Mechanisms and Nutritional Strategies to Prevent Kidney Failure’

With the new title: ‘CKD Diets for Kidney Failure Prevention: Insights from the IL-11 Paradigm’

Secondly, we have omitted the term "strategies" from the last paragraph of the Introduction, which outlines the scope of our review.

In general this manuscript is an extensive review on novel mechanisms and nutritional strategies for the management of CKD. The main problem of the manuscript, although it cointains novel end interesting information,  it needs more efford in order to have make more clrea the connection between diet and the mechanisms of kidney repair and regeneration. 

Thank you very much for your appreciation of our work. It lays the groundwork for future studies by us and others on identifying nutrients that influence regenerative processes.

The review has not followed the guidelines of scoping or systematic reviews. Therefore it is of limited value.

The journal "Nutrients" by MDPI offers various review formats, including the following

"Review: These provide concise and precise updates on the latest progress made in a given area of research. Systematic reviews should follow the PRISMA guidelines."

Our review contributes to the latest progress made in the field of CKD regenerative biology and maladaptive response, with a focus on advancements in the IL-11 pathway. We explored a segment of nutrition research examining IL-11 levels within the context of diseases relevant to CKD. Additionally, we discussed clinical trials involving traditional CKD diets to underscore their benefits, although they did not result in treatment or significant regression of CKD.

  1. No methodology of searching is presented and no evaluation of the quality of the selected stusied for the review is presented.

This was not intended to be a systematic review but rather to "provide concise and precise updates on the latest progress."

  1. The manuscript is so long that it looks as if two different reviews are mixed without a clear connection. Moreover, the paragraph on the enrichment is also a third thematic area, important but making the whole document exhaustive. 

We appreciate your comments. Manuscript was substantially shortened. We moved Table 1, 2, and a part of Table 3 into Supplementary material section as they are not representing the most novel findings.

4, Tables heading are too extensive and should be shortened. The relative info could be incorporated in the text. 

The table abbreviations are included in the text and are also provided in the 'Supplementary Materials.'

  1. The term elderly should be replaced by older adults in the text. 

The term "elderly" was replaced by "older adults."

  1. Figure 1 needs improvement. 

 Figure 1 is improved.

  1. I would see this manuscript more useful if the nutrition was more focused on the first part and the 1st part on kidney repair would me more focused on nutrition related aspects. Therefore I would suggest to modify all the figures and the text of 1st part providing more information on how diet can alter the mentioned pathways. 

Unfortunately, kidney regeneration and the IL-11 response are relatively new areas of research. As a result, the mechanisms by which nutrients can influence this pathway are just beginning to be explored, and the detailed mechanisms of these nutrients remain unknown to the extent of being presented as a specific Figure. Nonetheless, IL-11 has been measured as a marker of inflammation in several studies involving nutrients, and we have included this information in our review. In vitro studies also suggest specific mechanisms influenced directly or indirectly by nutrients. Therefore, we have added the potential site of interaction with nutrients to Figures 2 and 3. We hope that the information presented in this review will inspire further studies in this field.